# Addressing Human Factors in Cybersecurity Leadership

**William J. Triplett**

Department of Cybersecurity Leadership, Capitol Technology University, 11301 Springfield Road, Laurel, MD 20708, USA; wjtriplett@captechu.edu

**Abstract:** This article identifies human factors in workplaces that contribute to the challenges faced by cybersecurity leadership within organizations and discusses strategic communication, human–computer interaction, organizational factors, social environments, and security awareness training. Cybersecurity does not simply focus on information technology systems; it also considers how humans use information systems and susceptible actions leading to vulnerabilities. As cyber leaders begin to identify human behavior and processes and collaborate with individuals of the same mindset, an organization's strategy can improve substantially. Cybersecurity has been an expanding focal point from the viewpoint of human factors. Human inaccuracy can be unintentional due to an inaccurate strategic implementation or accurate unsatisfactory plan implementation. A systematic literature review was conducted to realize unintentional human factors in cybersecurity leadership. The results indicate that humans were the weakest link during the transmission of secure data. Furthermore, specific complacent and unintentional behaviors were observed, enabled by the ignorance of leaders and employees. Therefore, the enforcement of cybersecurity focuses on education, awareness, and communication. A research agenda is outlined, highlighting a further need for interdisciplinary research. This study adopts an original approach by viewing security from a human perspective and assessing how people can reduce cybersecurity incidents.

**Keywords:** cybersecurity; information technology; human factors; human behavior; leadership; security risks

## 1. Introduction

Today, cybersecurity leadership encounters enormous challenges in the work environment. Cybersecurity vulnerabilities have evolved into emergent threats for federal agencies or organizations over the last decade. To detect cybersecurity threats, organizations have spent billions of dollars on information technology (IT) systems and software [1]. The most substantial type of cybersecurity accountability is the management of individuals. Considering human factors in cybersecurity leadership is the key to a successful organization. Human factors comprise data elements, human behaviors, and human performance with an aim to reduce errors. Examples include human interactions with computer workstations and mobile devices, the unacceptable use of IT resources by employees, and hardware theft [2].

Disappointingly, leaders who view cybersecurity breaches as a technological catastrophe do not take on managerial responsibility in their organizations [3]. Cybersecurity entails the correlation between individuals and information systems; however, people frequently forget that cybersecurity issues require an understanding of human behavior. Schultz revealed the significance of the scarcity of cybersecurity analysts for human engineering and functional design, and summarized the value of recognizing how the organizational climate and business environment affect the application of well-informed workers that participate in the practice of internet security [3].

According to Schultz [3], leaders have failed to give proper attention to human behavior in their efforts to keep data research secure and to plan organizational strategies.

The evolution of cybersecurity vulnerability has highlighted the human dimension by producing depreciative outcomes for organizations, including insufficient IT investments, unsatisfied clients, economic losses, and substantial regulatory penalties [4,5]. Imprudent individuals utilize digital tools by sharing usernames and by distributing open data on social media platforms. The human element has historically been indicated as the most vulnerable spot in IT security [2,6]. Research has shown that humans are the most notable risk to security (86%), followed by the cybernetic sciences (63%) [7]. Senior management plays a critical role in this field, and humans should always be at the center of daily operations. It is often observed that human factors such as miscalculations cause more than 80% of cyber incidents, data violations, and malware attacks [8].

These data are also reflected in corporate communications surrounding cybersecurity and human behavior in the cybersecurity realm [8–14]. Corporate messaging has also missed the point when it comes to engaging citizens and improving behavior to ensure cybersecurity is protected, even though humans and human communication are critical to cybersecurity infrastructure protection [6]. As previously mentioned, given the importance of human action and human calculations to maintain secure cyber infrastructure, corporate communications must also be considered when developing a strategy to ensure cyber security moving forward [8].

## 2. Cybersecurity as a Citizen's Challenge

Cybersecurity is not solely a technological issue. At its core, cybersecurity is a sociotechnical issue because human factors are often the weakest link in creating a safe digital environment [9–14]. The leaking of sensitive information, whether intentionally or unintentionally, by employees who do not comply with their company's cybersecurity policy presents a serious issue. Organizations must plan detailed actions to scan technical and human vulnerabilities. Some cybersecurity breaches are linked to procedures such as password management. Understanding risks from a cultural perspective and at the enterprise level will aid in addressing the human element. Given the amount of information and passwords processed daily, this endeavor is conceivably exhausting. Individuals who experience employee burnout may forget their passwords. Authentication and passwords cause errors if they are too complicated for most users. Focusing on the user experience and security is important, as many human errors in cybersecurity are the result of employees and organizational factors. For example, leaders have found that a lack of security training, failure to enforce policies, absence of communication, overextension of employees, and workplace conditions contribute to stress [9,10]. Human performance is also influenced by the presence of managers and the expectations set by the leadership.

As leaders inquire into the elements that impact human behavior, individuals must always remember the role of the environment. The environment includes organizational factors, including the design and management structure, as well as the sociocultural context [10]. Organizations should establish a process to reduce human errors caused by a stressful work environment. Many companies are dealing with employee burnout and a shortage of cybersecurity staff. The condition of the workplace changes individuals' outcomes, given the idea that motivation can enhance human performance. A hectic corporate climate might generate adverse impacts on the whole organization, whereas appropriate management of stress can allow workers to reduce errors and can foster good cybersecurity behavior [6]. One of the vital concerns is the workers' devotion to business security policies, where a lack of awareness might cause substantial effects [2]. If someone is easily distracted, accidents in cybersecurity can occur. Security fatigue and stress are common causes of human errors. Organizations frequently identify security policies as simply another task or as an instrument applied by management to exert authority. To focus on policies and procedures, countless companies have built detailed security policies. Security policies are necessary for the success of organizations because they are part of the organizational culture. Simultaneously, as humans decide to act per mandates and security rules, they will follow through on cost-benefit strategies. Working closely with

people on security concerns enhances motivation; consequently, workers feel encouraged and engaged in critical operational strategies. The adoption of organizational guidelines helps even those workers who are aware of security issues. The security risks must be accessible to employees from the senior level down to the cybersecurity analyst.

## 3. Corporate Communications

People and cybersecurity are inseparable. From this perspective, communication plans are essential to cybersecurity leadership. For example, examining security policies in a committed board meeting or casual collaboration with other people has a distinctly positive effect [11–14]. Communication should include messaging designed to inform parties of the application of security policies. Instead of concentrating on the adverse after-effects of a communication breakdown, it is better to offer productive involvement. This approach can enrich an individual's optimistic awareness and strengthen human behavior. Communication is the key to security strategy and is necessary for every situation in which human beings work together. For organizations, communication plays an intentional role in cyber and human factors, both inside and outside the company, which allows leaders to ensure long-term success. Insufficient or fuzzy communication can impede effective cybersecurity. Limited communication within the IT organization will cause the managers to fail. The stresses to which leaders are subjected, alongside the obligation to focus on emerging threats, can force them to be detached from their coworkers [11–14]. When evaluating the outlook of the cybersecurity labor force, organizations must look beyond technology and engineering proficiencies [15], as well as consider the significance of linking communication and social abilities. Managers often have specialized skills, and agencies embrace personalities with diverse levels of expertise. Despite the ever-increasing digitalization of information systems, leaders cannot dismiss the importance of communication abilities for enhancing community involvement in the culture. Moreover, leaders can no longer overlook the value of the emotional components of an individual's receptiveness, particularly when mass media communication highlights the adverse impacts of cyber threats and information breaches [6].

## 4. The Role of Cybersecurity Leaders in Promoting Cybersecurity

Much of the research on cybersecurity in the workplace focuses on employees as both the most important source of vulnerability and the most important resource for defense [16–19]. Although individuals can make miscalculations, such as inserting thumb drives into computers and opening unauthorized emails, the responsibility for these problems does not rest on inattentive workers but instead lies with cybersecurity leaders who fail to address individual performance in the digital environment [20]. According to Parenty [1], the most effective defense is promoted through education and training, both of which fall under the responsibility of company leadership. Despite the importance of leaders in promoting cybersecurity, there is a notable gap in the literature on the role of these leaders as human factors supporting organizational cybersecurity. This is important because, for policies and training to be enforced, leaders must take accountability and guide training initiatives. This section explores the literature on cybersecurity leadership and makes the case for viewing cybersecurity leaders as human factors.

Several studies have examined cybersecurity governance and offered suggestions for making cybersecurity leadership more effective. Traditionally, the responsibility for cybersecurity falls to an organization's senior leadership team, including the CEO, COO, CFO, and CIO [21]. However, the involvement of so many people in cybersecurity leadership can make it difficult to clearly define leadership roles and hold leaders accountable. To help rectify this problem, the authors of [21] proposed that the responsibilities of a firm's chief information security officer (CISO) be expanded to make them a point person for all issues relating to cybersecurity. A CISO can strengthen a company's cybersecurity policy by promoting an encompassing risk management platform, developing an effective enterprise risk management strategy with a special focus on cyber risks, and employing

efficient communication regarding cybersecurity risk management and opportunities. For CISOs to be successful, they must be credible and can "communicate across the divide of senior management and engineering [but] be equally comfortable in the coat-and-tie boardroom as in the Hawaiian shirt-and-jeans back office and IT department" [22]. In other words, a successful cybersecurity leader can effectively interact with different levels of management. What is missing in this description, however, is the ability to successfully communicate with run-of-the-mill workers in a nonmanagerial position. Effective cybersecurity leaders must be able to quickly respond to cybersecurity threats and work with employees to develop and reinforce important cybersecurity skills. CISOs must, therefore, possess not only sharp technical skills but also business acumen and strong interpersonal and team-building skills.

Much of the literature on cybersecurity leadership focuses on relations occurring among senior leaders or on the importance of adopting new technological measures to prevent further cybersecurity incidents [21,23,24]. However, a focus on these issues precludes an awareness of the leadership skills required for implementing successful cybersecurity and how cybersecurity leaders themselves represent human factors in their organization. Indeed, cybersecurity leaders should view themselves as part of their company's entire business model and understand that their role is not limited to just security [25]. In an attempt to develop a novel cybersecurity leadership framework, Cleveland and Cleveland [26] argued that effective cybersecurity leaders should possess competence, a passion to help others, and mentorship. Additionally, they are capable of impacting their followers' strategies so that their leadership produces the desired response. Cybersecurity leaders should be able to incentivize their followers to adopt effective cybersecurity practices through their knowledge of and passion for such issues [27]. If workers see that leaders are implementing good cybersecurity practices and are committed to working with employees to develop best practices, they will be motivated to improve their practices. In this way, cybersecurity leaders can be considered human factors because their behavior arguably influences employees' cybersecurity behavior and an organization's cybersecurity performance.

Cybersecurity professionals tend to be underrepresented in organizations' leadership hierarchies [28]. This oversight not only makes cybersecurity policies harder to implement and enforce but also creates a distance between cybersecurity leaders and employees that hinders safe cyber practices. To effectively promote cybersecurity awareness, leaders need to have the social capital and competencies necessary to manage nontechnical employees [28]. Possessing technical expertise is not enough if a leader cannot communicate with subordinates and assume responsibility for making sure that employees are exerting appropriate cybersecurity practices. Rotherberger [29] claimed that cybersecurity leaders such as information technology managers and CISOs often lack the leadership experience and competencies to communicate and delegate cybersecurity awareness policies to employees. This observation is important because organizations continue to remain vulnerable to cyberattacks as long as leaders are unable to effectively communicate cybersecurity expectations and policies. Another cybersecurity leadership issue in many companies is that senior leaders are more concerned with engaging in technical operations than treating cybersecurity as a business strategy [30]. This means that cybersecurity leaders engage with executive-level functions at the expense of mentoring and training employees in cybersecurity policies.

## 5. Materials and Methods

This study employed a systematic literature review as the research method of choice. Munn et al. defined a systematic literature review as a critical appraisal of research journals and articles to analyze data or information needed for a particular problem or responses to a particular set of research questions [30]. Pati and Lorusso stated that a systematic literature review presents researchers with the opportunity to investigate the quality, levels, and amount of evidence that exist on a particular topic or phenomenon of

interest [31]. A systematic literature review provides scholars with a much broader understanding of the area of interest as the researcher dissects and analyzes each piece of evidence independently and minutely.

In this study, we sought evidence on the role of human factors in influencing cybersecurity leadership. Although research on human factors is broad, we narrowed down the area of interest to focus on unintentional human factors facilitating cybersecurity attacks in organizations and preventive measures adopted by organizational leaders. We followed the systematic literature review protocol to search for information that would help justify the existence of the identified problem. According to Pati and Lorusso, researchers conducting a systematic literature review should adhere to the following protocols or guidelines: searching for systematic relevant literature, filtering identified literature, reviewing selected studies, critiquing the literature, interpreting the literature, and reporting the findings from the different perspectives of the reviewed studies [31].

For this research, the first step was the identification of keywords that would facilitate the easy retrieval of information from different academic databases. The keywords used included cybersecurity, unintentional insider threats, leadership "AND" cybersecurity, human factors in cybersecurity leadership, cybersecurity, "AND" unintentional human factors. The use of search databases facilitated the literature search in various databases, including Springer, ScienceDirect, Emerald, IGI Global, IEEE explore, IDEAS, ACM Digital Library, and Google Scholar. However, most reviewed studies were obtained from the Web of Science (WoS) repositories. Given the enormous volume of research on cybersecurity and the role of human factors in cybersecurity, inclusion and exclusion criteria were adopted. As inclusion criteria, the studies included had to have been published in the last 5 years, i.e., from 2018 to date, the studies had to be focused on unintentional human factors in cybersecurity, the studies had to involve humans in both online and offline contexts, and the studies had to be qualitative, quantitative, or systematic literature reviews. Moreover, we excluded studies that were published before 2018, studies with a mixed-methods research design, and studies concentrated on cybersecurity systems rather than on humans. In addition, studies that considered autistic, deaf, or blind people were excluded.

The initial literature review yielded numerous studies on human factors and cybersecurity, some of which were irrelevant. After filtering and analysis, we systematically reviewed 15 qualitative systematic reviews and quantitative studies on unintentional human factors in cybersecurity leadership (Table 1). This study allowed for cumulative identification of human factors and underlying mechanisms, as well as of initiatives for enforcement and suggestions for future research and practice.

## 6. Results

A total of 15 studies were reviewed to explore how unintentional human factors influenced cybersecurity leadership. Table 1 shows that, out of the 15 reviewed studies, one employed quantitative methodology, whereas 10 employed qualitative research methods (either exploratory research design or multiple case studies); the remaining four were systematic literature reviews. The overall findings were that, compared with intentional insider threats, unintentional threats associated with human factors were just as costly. The main unintentional factors identified included forgetting to log out of a computer system and unknowingly clicking fraud emails and links due to limited knowledge and skills in cybersecurity. Although employee training was a common mechanism identified for cybersecurity enforcement, several studies suggested alternative mechanisms. Table 2 indicates citation data for the most cited articles selected. The most cited article was Zwilling et al., which was also the most recent [32].

**Table 1.** Included studies.

| Citation | Repository | Purpose | Participants | Method | Findings | Enforcement | Suggestions |
|---|---|---|---|---|---|---|---|
| Aldawood and Skinner [33] | WoS | To raise awareness and educate employees on cybersecurity social engineering | Six cybersecurity specialists; 15 articles | Qualitative | Defines social engineering as manipulating a user of technology by deceiving them. Identifies humans as the weakest link in organizational security. | Education in cyberattacks helped reduce incidences. | Qualitative research is needed for employee understanding of cybersecurity. |
| Nobles [34] | IDEAS/ RePEc | To explore human factors influencing cybersecurity in organizations | Nine participants | Qualitative | Cyber-related attacks are propagated by human factors. Nevertheless, managers were reluctant to equip themselves with the knowledge, skills, and expertise to effectively mitigate cyberattacks. | Found lack of cybersecurity training or other enforcements. | Employees, as well as college and university students, should be trained in human factors associated with cybersecurity. |
| Dawson and Thomson [15] | WoS | To review the literature on the future of cybersecurity in the workforce | | Systematic literature review | The analysis revealed six themes, including team players, sense of civic duty, social skill, and technical skills, which would be critical in addressing cybersecurity issues. | n/a | Research should examine the cognitive underpinnings of intentional and unintentional cybersecurity risks. |
| Wong et al. [35] | WoS | To explore the human factors behind information leakage and the mitigation of insider threats | Five managers from five companies | Qualitative multi-case studies | Information leakage occurred because of intentional and unintentional human behavior. | Mitigation of leakage includes clear ethical codes enforced by an ethical organizational climate and employee training. | Future research should consider quantitative analysis and extension of the geographical reach of studies. |
| Jeong et al. [9] | IEEE explore | To provide an understanding of human factors in cybersecurity | 27 articles | Systematic literature review | Personality, demographic, and cultural contexts influence employee behavior to unintentionally facilitate malicious attacks. | n/a | Cybersecurity research should incorporate findings from other fields regarding the impact of human factors on technology. |
| Ani et al. [16] | Emerald | To evaluate the human factor in industrial cybersecurity efforts | 37 cybersecurity specialists | Quantitative | Lack of knowledge and skills on cybersecurity, negligence, and misinformation on cybercrimes may unintentionally spur increased cases of cyberthreats. | Unintentional effects were mitigated by making cybersecurity training intentional and mandatory. | Future studies can develop automated evaluation tools with cognitive and behavioral components for understanding human factors. |

| | | | | | | | |
|---|---|---|---|---|---|---|---|
| Williams et al. [36] | Springer | To explore human error in information security, specifically multitasking and interruptions | 15 participants | Qualitative | Distraction of employees by unplanned interruptions and multitasking unintentionally facilitates cyberattacks. | n/a | Continued empirical research in cyberpsychology to guide human–machine solutions for cybersecurity issues. |
| Maalem Lahcen et al. [37] | WoS | To explore the role of social and behavioral aspects of cybersecurity | | Systematic literature review | Human factors facilitating cyberattacks include lack of communication, distractions, lack of teamwork, lack of knowledge and skills, and complacency. These factors facilitated unintentional errors and increase organizations' vulnerability to attacks. | Education as a preventive measure. | An interdisciplinary conceptual framework is needed to investigate behavioral cybersecurity, human factors, modeling, and simulation. |
| Kadena and Gupi [38] | Google Scholar | To explore human factors in cybersecurity with the associated risks and factors | | Literature review | Inadequate use of technology by employees, the management's lack of motivation, and inadequate staffing expose organizations to cyberattacks. | n/a | Private and public cybersecurity companies should be considered in cybersecurity studies. |
| Abulencia [39] | Science Direct | To understand insider attacks from the perspective of human factors and mitigation | | Conceptual | Unintentional human factors such as miscommunication, forgetting company policies and procedures, and limited skills and information on cyberattacks may contribute to increased incidents. | n/a | A holistic approach to cybersecurity should be applied instead of analyzing one risk at a time. |
| Nifakos et al. [40] | WoS | To investigate how human factors impact cyber security in healthcare organizations | 70 articles | Systematic literature review | Many cyberattacks exploited human weakness, including ignorance of cyber threats to healthcare employees and management. | n/a | There is a need to evaluate the effectiveness of training employees on human factors. |
| Rahman et al. [41] | ACM Digital Library | To investigate the role of human factors in cybersecurity | 27 studies | Systematic and scoping literature review | Employees' and leaders' social influence, attitude, feelings of usefulness, and perceptions of security impacted their use of technology and the likelihood of being cyberattacked. Related skills and a positive attitude on the use of technology protect against cyberattacks. | Training in cybersecurity skills can reduce cyberattacks. | A qualitative grounded theory research method focusing on the influence of culture could improve research on human factors in cybersecurity. |

| Randall and Allen [42] | WoS | To explore how cybersecurity professionals share information in the electrical sector | 13 participants from 10 organizations | Qualitative exploratory case study | Sharing of information exists at interpersonal and intergroup levels. | The impacts of human factors could be addressed via law enforcement agencies and the development of critical infrastructure. | There is a need to examine infrastructural factors that enhance human factors in promoting cyberthreats. |
|---|---|---|---|---|---|---|---|
| Georgiadou et al. [43] | WoS | To investigate how a cybersecurity culture framework can help detect insider threats | 28 IT employees and 449 non-IT employees | Qualitative multi-case study research design | The cybersecurity culture framework helps to prevent human behaviors that would facilitate unintentional attacks on organizational information systems. | Appropriate cultural norms help enforce cybersecurity. | The study could be extended to diverse work responsibilities, more dimensions of the cybersecurity culture framework, and wider geographical coverage. |
| Hadlington [44] | IGI Global | To explore the human factor in cybersecurity | | Literature review | Cybersecurity research has focused on the role of disgruntled and greedy employees facilitating malicious attacks, but there is inadequate research on unintentional factors, such as poor planning, ignorance, and lack of attention. | Behavioral nudges were identified as an enforcement mechanism of cybersecurity. | Theory on behavioral nudges could contribute to this field. |
| Ramlo and Nicholas [45] | WoS | To assess the individual perception of cybersecurity and the diverse views regarding cybersecurity | Six semi-structured interviews | Qualitative | Four perspectives that relate to individual perceptions of cybersecurity were identified, namely, best practices, poor cybersecurity behaviors such as worried but not vigilant persons, naïve cybersecurity practitioners, and cybersecurity as a big problem. | The implementation of best practices could improve cybersecurity. | More research on cybersecurity best practices is needed with regard to human factors. |
| Zwilling et al. [32] | WoS | To examine cybersecurity in terms of awareness, knowledge, and behavior | 459 participants from Israel, Poland, and Turkey | Quantitative regression analysis | Internet users were well aware of cyberthreats but employed limited protective measures. Cyber connectedness is closely associated with cybersecurity awareness. | A lack of cybersecurity training was identified. | Future research could explore training programs to increase cyber knowledge, awareness, and connectedness. |

**Table 2.** The most cited studies in the sample, with citation information.

| Year | Authors | Journal/Conference | Title | Citations in WoS | Citations in Google Scholar |
|---|---|---|---|---|---|
| 2018 | Dawson and Thomson | Journal | The future cybersecurity workforce: going beyond technical skills for successful cyber performance | 29 | 115 |
| 2022 | Zwilling et al. | Journal | Cyber security awareness, knowledge and behavior: a comparative study | 24 | 67 |
| 2018 | Aldawood and Skinner | Conference | Educating and raising awareness on cyber security social engineering: a literature review | 19 | 70 |
| 2021 | Hadlington | Journal | The "human factor" in cybersecurity: exploring the accidental insider | n/a | 56 |
| 2019 | Ani et al. | Journal | Human factor security: evaluating the cybersecurity capacity of the industrial workforce | n/a | 44 |
| 2018 | Nobles | Journal | Botching human factors in cybersecurity in business organizations | n/a | 29 |
| 2019 | Wong et al. | Journal | Human factors in information leakage: mitigation strategies for information sharing integrity | 9 | 21 |
| 2021 | Georgiadou et al. | Journal | Detecting insider threat via a cybersecurity culture framework | 4 | 5 |
| 2020 | Maalem Lahcen et al. | Journal | Review and insight on the behavioral aspects of cybersecurity | 2 | 15 |
| 2019 | Jeong et al. | Conference | Toward an improved understanding of human factors in cybersecurity | n/a | 12 |

## 7. Discussion

The results of the systematic literature review revealed that humans are the weakest link in the increased cases of cyberattacks. The discussion provided by Nobles [34], Dawson and Thomson [15], and Randall and Allen [42] confirmed that hackers and perpetrators focused on the slip of employees and support staff to facilitate malicious attacks against an organization. For instance, Rahman et al. [41] and Noble [35] explained that 95% of malware and ransomware attacks were perpetrated by humans, either intentionally or unintentionally. Noble [35], together with Ramlo and Nicholas [45] and Wong et al. [35], revealed that employees who felt underappreciated or vindictive in their work for one reason or another facilitated intentional cyberattacks on the organization. Wong et al. [35] contended that harboring negative thoughts toward an organization and the need to make financial gain from leaking classified information explained increased cases of intentional insider cyberattacks. Similar findings were reported by Hadlington [44], who opined that disgruntled and greedy employees were likely to facilitate insider intentional attacks on the organization.

In addition to intentional cyberattacks, Hadlington [44] noted enormous existing research focused on the intentional human behavioral factors that facilitated cyberattacks on the organization. Agreeing with Hadlington [44], Maalem Lahcen et al. [37] asserted that intentional distractions and multitasking exposed the vulnerability of human beings in cyberspace, thereby increasing cases of cyberattacks. Kadena and Gupi [39], in an extensive literature review, explained that lack of motivation in the management or organization's leadership and employees' limited use of technology facilitated increased cyberattacks. Kadena and Gupi [38] explained that intentional human behaviors, including intentionally failing to follow the laid down procedures and protocols and disgruntlement among employees for reasons such as inadequate staffing, encouraged employees to engage in harmful behaviors that might expose an organization to cyberattacks.

Although many scholars concentrated on intentional insider human factors in promoting cyberattacks, recent scholars have shown an increase in unintentional human factors that promote cyberattacks. In the above-reviewed studies, most qualitative studies showed that unintentional human factors increased cyberattacks on organizations. For instance, Kadena and Gupi [38] established that organizational management facilitated most unintentional attacks on the organizations' information systems. For instance, most of those in leadership positions did little to promote and encourage the use of technology among their employees. Aldawood and Skinner [33] established that limited use of technology by employees limited the ability to recognize deception used by hackers in social engineering. Aldawood and Skinner [33] and Kadena and Gupi [38] presented that inexperienced employees were likely to be deceived and unknowingly click on links and navigate websites that allow hackers to gain entry into organizations' systems without them knowing.

Similar findings were reported by Rahman et al. [41], Ani et al. [16], and Nifakos et al. [40]. Rahman et al. [41] asserted that, due to limited technological skills, many organizational employees leave without logging out of their systems or create weak passwords that could be cracked easily by hackers. Leaving laptops and computers open presents leeway for other employees with malicious intent to plant malware or leak critical information. Ani et al. [16] used a qualitative research method to study human factors as facilitators of cyberattacks and established that the lack of knowledge and skills in cybersecurity resulted in employees failing to recognize critical red flags of cyber breaches. In addition, they explained that increased cases of negligence coupled with misinformation spurred an increase in cyber threats as both employees and leaders could not recognize that they were hacked or had provided easy access to their technological systems. Nifakos et al. [40] corroborated the findings of Ani et al. [16] and Rahman et al. [41], which concluded that many organizational leaders, despite knowing the relevance of being cyber secure, were still ignorant of the severity and damage cyberattacks would cause to the

organization. Reference to humans as the weakest link for increased cases of cyberattacks is due to the ignorance portrayed by leaders and employees.

Expanding the findings on employee and leadership ignorance, Abulencia [39], Nobles [35], and Ramlo and Nicholas [45] built their argument on the complacency of organizational leadership, which is likely to be adopted by employees. Ramlo and Nicholas [45] identified that poor cybersecurity practices and naïveté on the part of users of technology explained how complacent organizations were increasingly made to be the victims of cyberattacks. Moreover, Nobles [34] argued that managers were reluctant to equip themselves with the necessary knowledge on cybersecurity or even seek the help of cybersecurity experts, making them complacent and slow in responding to cyberattacks. Maalem Lahcen et al. [37] asserted that poor communication among employees, limited teamwork, and unintended disruptions encouraged complacency, which increased cases and rates of cyberattacks.

The enforcement of cybersecurity was found to require more training by seven of the studied articles [16,33–36,38,42]. In particular, Ani et al. [16] explained that making cybersecurity lessons and training mandatory would help protect against unintentional data breaches by increasing leaders and employees with the skills to be careful around data and recognize cyberattacks easily. Rahman et al. [41] also found that possessing the required skill set and a positive attitude toward the use of technology helped reduce unintentional data breaches. Maalem Lahcen et al. [37] and Aldawood and Skinner [33] discussed that promoting innovative education on cybersecurity not only enhanced the awareness of cyberattacks and cybercrimes but also promoted setting passwords that were strong but easy to remember. Furthermore, increasing employee and leader attention regarding the use of technology and innovative education enabled users of technology to log out each time they were through with activities in order to prevent authorized access. Education would also help leaders to address issues with fatigue by promoting the use of technology, motivating their employees to hone skills in technology, and using the same technology to address intentional disruptions and complacency. In addition to training measures, the role of organizational climate or culture in enforcing cybersecurity was identified by Wong et al. [35] and Georgiadou et al. [43]. Randall and Allen [42] proposed alternative enforcement measures, i.e., law enforcement agencies and the development of infrastructure, whereas Hadlington [44] considered enforcement through behavioral nudges, and Ramlo and Nicholas [45] suggested the implementation of best practices.

The suggestions for future research of the studied articles outline a research agenda for human factors in cybersecurity. The call for interdisciplinary studies drawing on related fields was reflected in several studies [9,38]. In this context, Dawson and Thomson [15] and Ani et al. [16] suggested research based on cognitive theory. Williams et al. [36] suggested using psychological theory, whereas Hadlington [44] highlighted the benefits of recent advances in behavioral theory.

The topics visited in the discussion are summarized in Table 3.

**Table 3.** A discussion summary.

| Discussion Topics | Notes |
|---|---|
| Humans impede cybersecurity | Humans are the weakest link of cybersecurity. Unintentional activities include setting weak passwords and forgetting to log out of computer systems. |
| Complacency | Organizational leaders are complacent or unintentional in instituting policies and measures that would protect organizations from cyberattacks. |
| Ignorance | Leaders and employees are ignorant of the red flags and links marked as suspicious. |
| Enforcement | Organizations have become reluctant in training employees on cybersecurity, increasing the organizations' vulnerability, as illustrated by forgetting to log out of computer systems and setting up weak passwords that are easy to crack and infiltrate. |
| Interdisciplinarity | Future research in cybersecurity would benefit from analyzing human factors using behavioral and cognitive theories. |

Given that leaders are an often-overlooked human factor influencing organizations' vulnerability to cyberattacks, several practice recommendations can be made. First, the accountability of cybersecurity leaders in mitigating risk and preventing organizational exposure to cybersecurity risk needs to be improved. One way of creating accountability is by creating a cybersecurity charter, signed by cybersecurity leadership and other members of the company's executive leadership, in which all leaders agree to not expose the organization to risk. Second, cybersecurity leaders themselves should lead the charge in developing goals and indicators for cybersecurity, as well as support workers in meeting these targets. In this sense, leaders should employ a more hands-on approach to cybersecurity that focuses on transforming employee behavior by coaching them to success, monitoring their progress, and helping them understand the cybersecurity strategy [26]. To bridge the gap between leaders and employees, the former should also help the latter adapt to change by working together to better integrate employees into the organization's new cybersecurity framework [26]. Third, organizations need to implement leadership development programs to better prepare their cybersecurity leaders to work with other employees. Developing cybersecurity leaders is an investment, and the return on this investment will take the form of reduced cybersecurity risk [28]. Such leadership development programs should include activities that promote team building, self-awareness, emotional intelligence, and trust, all of which are crucial to improving managerial and decision-making skills [28].

## 8. Conclusions

To summarize the Discussion section, the findings of this study contribute to the following domains explored:

1.  A cumulative analysis of specific human factors in cybersecurity leadership, including complacent or unintentional behaviors;
2.  An analysis of the underlying mechanisms, highlighting the ignorance of leaders and employees;
3.  A cumulative analysis of enforcement initiatives focused on training and including alternative behavioral, cultural, and infrastructural measures;
4.  A research agenda identifying the recurrent suggestions for future research regarding human factors in cybersecurity, highlighting the usefulness of behavioral and cognitive theories.

Cybersecurity leaders play a critical, if overlooked, role in promoting organizational cybersecurity at the employee level. These leaders are human factors that contribute to the level of vulnerability that an organization faces. Human factors consist of human behaviors and human performance to reduce errors [20]. Responsibility for these problems does not rest on inattentive workers but instead lies with cybersecurity leaders who fail to address individual performance in the digital environment [20]. Since the onset of the COVID-19 pandemic, working from home has become the new norm, but cybersecurity has also become a bigger issue that organizations need to address because workers often use their own devices to download software and access company data. A lack of awareness of cybersecurity by workers can cause substantial effects such as when they are easily distracted, stressed, and fatigued, whereby security accidents can occur [6]. However, blame cannot be attributed to workers alone. Cybersecurity leaders have a responsibility to ensure that company policies are being followed, and they need to be held accountable for enforcing cybersecurity policies. Furthermore, leaders need to be better equipped to effectively communicate with workers regarding these issues [26,28]. By involving workers, leaders can inspire the enrichment of optimistic awareness and strengthen human behaviors concerning cybersecurity [11–14].

**Funding:** This research received no external funding.

**Data Availability Statement:** Not applicable.

**Acknowledgments:** I express my deep appreciation to Ian, my research colleague, for his patient guidance, inspiration, and helpful assessments of this cybersecurity work. I also thank Nobles, Burrell, and Muller for their advice and support in keeping me on schedule. My grateful thanks are also extended to Jo for his help with conducting the data analysis. His willingness to give his time so generously is very much appreciated. Lastly, I thank my wife and parents for their support and encouragement.

**Conflicts of Interest:** The author declares no conflict of interest.

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
