# Peer review of "Addressing Human Factors in Cybersecurity Leadership"

_jcp, doi:10.3390/jcp2030029_

Round 1

Reviewer 1 Report

This paper presents the important issue of human factors in cybersecurity. The human is always the weakest link in a cybersecurity chain and can be phished, tricked or misled into giving a potential attacker access.

The conclusions in the paper are correct however I find them not very novel as this issue is well known. It is standard in larger organizations to educate employees in cybersecurity issues. Also, all the efforts to get rid of password-based systems is to large extent fuelled by the aspect of human factors.

So while the paper presents the right conclusions and some good references for that, the content and the conclusions are not very novel.

I suggest to make a more extensive study in order to make this a publishable paper. It is rather easy to search for a survey like that and find some with significantly more references

Author Response

Manuscript Titled: “Addressing Human Factors in Cybersecurity Leadership”

Manuscript ID: jcp-1717402

Journal: Cyber Security and Privacy

Reviewer 1:

Author response:

Thank you for this insightful comment. I have deeply analyzed the human factors and underlying mechanisms and have provided initiatives for enforcement and suggestions for future research and practice. Therefore, the main research contributions are clearer and are now listed in the first paragraph of the conclusion section.

“To summarize the discussion section, the findings of this study contribute to the explored domain as follows.

  1. A cumulative analysis of specific human factors in cybersecurity leadership, including complacent or unintentional behaviors. This paper presents the important issue of human factors in cybersecurity. The human can be considered the weakest link in a cybersecurity chain and can be phished, tricked, or misled into giving a potential attacker access. The role of leaders was not presented in the existing literature although we know that humans are the “weakest link”, to-date literature has focused on addressing issues at the employee-level (via training, education) and on mitigation of cybersecurity risks and breaches, and the role of cybersecurity leaders in the organization.
  2. An analysis of the conclusions of the the paper was conducted. Ignorance of leadership is an exception in relation to the role of leaders within cybersecurity this is noted within the conclusion. Lack of attention to the role of cybersecurity leaders as being responsible for addressing issues at the employee-level. Also, all the effort between the distance in proximity between employees and cybersecurity leaders.    
  3. A cumulative analysis of references and conclusions were added to enforce the focus of the leader within cybersecurity. Lack of emphasis on the role of cybersecurity leaders within the literature is reflective of the issue in practice in which less attention is given to the role of leaders in successfully implementing interventions to address the human factors that contribute to cybersecurity issues within the organization.

  1. A research agenda identifying the recurrent suggestions for future research regarding human factors in cybersecurity leadership and human factors. Therefore, I hope you will find the contribution of the paper clearer in the revised version with additional references and agree that it constitutes an advance in cybersecurity.

Reviewer 2 Report

Although the reasoning and the research method appear sound, I find the overall contribution to be lacking. The key points of this paper can easily be found on most of the web articles dealing with the human element of the security. We know that weak passwords are the problem. We know that lack of education and security awareness is a problem. And so on.

There are also some concerning statements, such as: "Cybersecurity is a citizen’s challenge, not a technological issue". Stating that cybersecurity is not a technological issue is troublesome to say the least, no matter how hard one tries to emphasize the importance of the human factor.

The writing is often unfocused and at times difficult to follow, bordering on the essay, with plenty of redundancy.

I would recommend a major revision which deep-dives into the topic with significantly more focus and conciseness, and a more substantial contribution.

Author Response

Reviewer 2: Comments and Suggestions for Author

Manuscript Titled: “Addressing Human Factors in Cybersecurity Leadership”

Manuscript ID: jcp-1717402

Journal: Cyber Security and Privacy

  1. Cybersecurity leaders play a key role in implementing interventions such as education, training, etc. to mitigate/address cybersecurity weaknesses within organizations, yet, as reflected within the literature review, there is limited focus on the role of these leaders as human factors supporting cybersecurity within organizations. Note: highlight the limited literature (i.e., ignorance of leaders, etc.) as the literature that demonstrates the absence of the role of cybersecurity leaders in literature and therefore in academic discussions.  The role of cybersecurity leaders in identifying the gap that is causing the cybersecurity risks and breaches (training, education, etc.). However, enforcement is needed.
  2. Cybersecurity is not solely a technological issue. At its core, cybersecurity is a sociotechnical issue because human factors are often the weakest link in creating a safe digital environment. The leaking of sensitive information, whether intentionally or unintentionally, by employees who do not comply with their company’s cybersecurity policy presents a serious issue. Emphasize that although we know that weak passwords, phishing emails, and other human factors are a primary cause of cybersecurity breaches. In order for policies and procedures to be enforced, leaders must take accountability and guide training initiatives
  3. Focused on updating the writing and defining accountability by creating a cybersecurity charter, signed by the board and cybersecurity leadership to agree that leaders will not expose the organization to risk. Security leaders should lead the charge in developing goals and indicators for cybersecurity and support meeting these targets. Need for active leadership intervention. Emotional intelligence and soft skills needed among cybersecurity leaders to implement, enforce, and monitor the issue at the employee level
  4. The concept of human factors (as expressed by the reviewers) – the focus should instead focus on emphasizing the key role of cybersecurity leaders yet the absence of research on the role of cybersecurity leaders in addressing the issues at the employee-level
  • Cybersecurity leaders play a key role in implementing interventions such as education, training, etc. to mitigate/address cybersecurity weaknesses within organizations, yet, as reflected within the literature review, there is limited focus on the role of these leaders as human factors supporting cybersecurity within organizations.
    • Note: highlight the limited literature (i.e., ignorance of leaders, etc.) as the literature that demonstrates the absence of the role of cybersecurity leaders in literature and therefore in academic discussions
  • Recommendations for future research:
    • Focus on the role of cybersecurity leaders in addressing cybersecurity issues at the employee-level
    • Given that human factors are the “weakest link” – there needs to be a consideration of leaders as those accountable/responsible for overseeing employees and mitigation potential risks related to such human factors
    • Causes for challenges in interventions (education, communication, training, etc.) implemented by cybersecurity leaders
    • Cyber risk in work culture as an enabler – how is cyber risk within the work culture addressed and the role of leadership
  • Recommendations for practice
    • Change management; end-user adoption of cybersecurity practice at the employee-level
    • Improving accountability of cybersecurity leaders to mitigate risk and prevent organizational exposure to cybersecurity risks
    • Practical recommendation:
      • Define accountability by creating a cybersecurity charter, signed by the board and cybersecurity leadership to agree that leaders will not expose the organization to risk
      • Security leaders should lead the charge in developing goals and indicators for cybersecurity – and support meeting these targets
      • Need for active leadership intervention

Round 2

Reviewer 1 Report

The improved version of the paper brings in an additional focus on leaders and their connection to cybersecurity issues. This is an important topic that is actually often coming in short.

Author Response

Author response:

Thank you for this insightful comment. I have thoroughly analyzed the human factors and underlying mechanisms and have provided initiatives for cyber law enforcement and suggestions, which will aid in future studies and practices. The main contributions of this study are clearer and are listed in the first paragraph of the conclusion section. The improved version of the article ties in the focus of the leaders regarding cybersecurity issues

“To summarize the discussion section, the findings of this study contribute to the explored domain as follows:

  1. A cumulative analysis of specific human factors in cybersecurity leadership, including complacent or unintentional behaviors.
  2. An analysis of the underlying mechanisms, highlighting the ignorance of leaders and employees.
  3. A cumulative analysis of enforcement initiatives focused on training and including alternative behavioral, cultural, and infrastructural measures.
  4. A research agenda identifying the recurrent suggestions for future research regarding human factors in cybersecurity, highlighting the usefulness of behavioral and cognitive theories.”

I hope that the revisions address your query as well as clearly indicate the contribution of the present study and agree that it constitutes advancement in cybersecurity.

Reviewer 2 Report

More text and references are added in this revision, but the included  recommendations can be found throughout the publicly available literature and industry standards, and I don't see them as an added contribution. The authors should have kept the focus on the literature review, which is done solidly. My final recommendation is to keep focus and remove everything that is not directly related to the research.

Author Response

Response to Reviewer 2 Comments

Point 2:

Author response:

Thank you for this insightful comment. I have thoroughly analyzed the human factors and underlying mechanisms and have provided initiatives for cyber law enforcement and suggestions, which will aid in future studies and practices. The main contributions of this study are clearer and are listed in the first paragraph of the conclusion section.  The cover letter addresses the updates regarding the the literature review, and keep the related research focused.

“To summarize the discussion section, the findings of this study contribute to the explored domain as follows:

  1. A cumulative analysis of specific human factors in cybersecurity leadership, including complacent or unintentional behaviors.
  2. An analysis of the underlying mechanisms, highlighting the ignorance of leaders and employees.
  3. A cumulative analysis of enforcement initiatives focused on training and including alternative behavioral, cultural, and infrastructural measures.
  4. A research agenda identifying the recurrent suggestions for future research regarding human factors in cybersecurity, highlighting the usefulness of behavioral and cognitive theories.”

I hope that the revisions address your query as well as clearly indicate the contribution of the present study and agree that it constitutes advancement in cybersecurity.

This manuscript is a resubmission of an earlier submission. The following is a list of the peer review reports and author responses from that submission.

Round 1

Reviewer 1 Report

The research contribution is not very significant, other than summarizing 15 papers and concluding that humans are still the weakest link, which is already an acknowledged fact based on obseravtions and empirical studies. In order to systematically study human factors posed as cyber risks, should analyze the MITRE ATT&CK Framework TTPs and systematically create the tactics and techniques from the human factor perspective that lead to initial stages of the attack vectors.

There is some font mismatch in some sections.

No rationale was provided why and how the 15 studies were chosen. 

Author Response

1.The research contribution is not very significant, other than summarizing 15 papers and concluding that humans are still the weakest link, which is already an acknowledged fact based on observations and empirical studies.

Cyber security has been an expanding focal point within the field of human factors. Human in-accuracy can be unintended due to the inaccurate implementation of a strategy or accurate im-plementation of an unsatisfactory plan. A systematic review of literature was used to study unin-tentional human factors in cybersecurity leadership. The results of the review were such that in-dividuals continue to be the weakest component in the process between the computer and the transmission of secure data. I found that I needed to conduct more research to make my points more clear and valid. Also, I summarized the limitations, suggestions, future work, and participants within the the updated studies. It will make it easier for me as the researcher to reach my audience to clearly define and state my postion contributing to human factors and empirical studies.

2.In order to systematically study human factors posed as cyber risks, should analyze the MITRE ATT&CK Framework TTPs and systematically create the tactics and techniques from the human factor perspective that lead to initial stages of the attack vectors.

Human factors posed as a cyber risks with limited access to tech support as in office, many individuals working from home were forced to access technological apps and programs by themselves in-creasing the risk of leaking company data or providing unintentional access to company servers using downloaded programs . While, systematic review of literature presents an opportunity to explore the area under study from a wider perspective, it is limited with time and accuracy of pre-sented data and size of the sample used. However, the findings provided in this study sets ground for future work on human factors and how they facilitate unintentional cyberthreats and cyberattacks. For future work, re-searchers should consider qualitative research to understand unintentional cyber-attacks and human factors from the perspectives and views of participants.

3.There is some font mismatch in some sections.

The mismatch font was aligned according to the Jornal of Cybesecurity and Privacy guidelines. Authors must use the Microsoft Word template or LaTeX template to prepare their manuscript. Using the template file will substantially shorten the time to complete copy-editing and publication of accepted manuscripts. The total amount of data for all files must not exceed 120 MB. If this is a problem, please contact the Editorial Office [email protected]. Accepted file formats are:

4.No rationale was provided why and how the 15 studies were chosen. 

The current rationale employed a systematic review of literature and table findings that explored how the different human factors influenced digital security of organizations. While, systematic review of literature presents an opportunity to explore the area under study from a wider perspective, it is limited with time and accuracy of pre-sented data and size of the sample used. One common limitation associated with systematic review is that it is not easy to verify the authenticity and accuracy of information and results provided. The second limitation is that while it is an easy research method, it is highly time consuming and tedious. However, the findings provided in this study sets ground for future work on human factors and how they facilitate unintentional cyberthreats and cyberattacks. For future work, re-searchers should consider qualitative research to understand unintentional cyber-attacks and human factors from the perspectives and views of participants. Recommended the examination of human factors using grounded theory of qualitative research method with a focus on the influ-ence of cultural works on human factors to develop a theory that can guide future research on cybersecurity.

Reviewer 2 Report

The aim of the study is providing a systematic review of literature to study unintentional human factors in cybersecurity leadership. In general, I enjoyed reading the manuscript and I think that the topic is of important especially from the practical point of view. The quality of the manuscript is below average. In most parts the manuscript is well written but there is no uniformity in size of fonts which makes it harder for reading. I also don’t find that the findings are sufficient for the acceptance and publication of the manuscript.

Major Comments:

  1. The authors need to organize the findings in a much clear manner. I suggest adding another column to state the number of participants (respondents) if possible, as well as additional column after findings for suggestions of improvement of the findings (if suggested) by the authors.
  2. It is suggested to summarize the different topics provided in the discussion section in a separate table and to present the main highlights in the discussion section with a text. This will make it easy for the reader to follow and not to read again the same observations as appear in the Table.
  3. With respect to the 15 studies, it is suggested to focus mainly on the WoS (Web of science) repository. Please include this repository in the materials and methods section.
  4. The conclusion part is not sufficient and should be elaborated. I suggest including a paragraph, which provides insights about the findings if they are also mentioned as derived by a stressful environment, as this was mentioned in the introduction part, but not being addresses in the table. Most of the cases are described as those occurred by unsatisfied employees from their position or attitude.
  5. In the column of findings please remain only findings and exclude methods such as for example (describing the 13 participants from the 10 organizations [Randall & Allen, 2021]).
  6. Please provide a contribution part of the findings to the explored domain as part of the discussion or as a separate part.

            Minor Comments:

  1. Keep on font size to be uniformity. In some places it is small in some it is bigger.
  2. Please move chapter 7 (Materials and Method) before section 5 (Results).
  3. There are several internal errors such as: in P.10 you mention “Published 5 years” as an inclusion criterion for documents but an exclusion of studies that were published below in 2018 and below. Papers from 2018 are still relevant for the 5 years …
  4. Some of the references in the introduction are very old (such as Schultz, 2005) it is better including articles mainly from the past 5 years.
  5. I suggest showing the most prominent papers in the explored domain that are evaluated in the table and include the following details with respect to them in a separate table: Year of publication, Authors, Journal (J) or Conference (C), Title, #of citations.
  6. The discussion doesn’t address the issue of organizations with or without security policy and enforcement of regulations (in case the firm is regulated). It is suggested to add a column in the table and include in the discussion an attitude related to enforcement and enforcement of regulations by the firm as related to awareness of employees to cyber security if applicable.
  7. I Would like to see additional ref in P.10 as related to the materials and methods section. There is a lot of text with hardly any references to support it.
  8. It is suggested to improve the English writing style.
  9. Please sort the year of scanned papers in the table from the oldest to the newest.
  10. Please include the following manuscript in your references and the table: “Zwilling, M., Klien, G., Lesjak, D., Wiechetek, Ł., Cetin, F., & Basim, H. N. (2022). Cyber security awareness, knowledge and behavior: A comparative study. Journal of Computer Information Systems, 62(1), 82-97.‏“
  11. Finally, I would like to see a future studies and limitations section in the manuscript.
  12. I believe that if the recommendations for the improvement of this manuscript will be taken point by point it will strengthen the quality and contribution of the manuscript.

Reviewer 3 Report

The article analyzes the human factor in leadership in cybersecurity. The research was based on a systematic literature review. In the opinion of the reviewer, the study can be supplemented with the following elements:

- The introduction should clearly state the purpose of the research.

- In the paper, it is worth adopting the criteria for assessing leadership in the field of cybersecurity. For example, what are the cybersecurity leadership requirements against which to be assessed?

- The article should include future research directions and/or recommendations for the improvement of cybersecurity leadership.

Author Response

The article analyzes the human factor in leadership in cybersecurity. The research was based on a systematic literature review. In the opinion of the reviewer, the study can be supplemented with the following elements:

1.The introduction should clearly state the purpose of the research.

Cyber security has been an expanding focal point within the field of human factors. Human in-accuracy can be unintended due to the inaccurate implementation of a strategy or accurate im-plementation of an unsatisfactory plan. The most substantial type of cybersecurity accountability is the management of individuals. Considering human factors in cybersecurity leadership is the key to a successful or-ganization. Human factors consist of data elements, human behaviors, and human per-formance to reduce errors. Examples include human interactions with computer work-stations and mobile devices, the unacceptable use of IT resources by employees, and hardware theft (Pollini et al., 2021).

2.In the paper, it is worth adopting the criteria for assessing leadership in the field of cybersecurity. For example, what are the cybersecurity leadership requirements against which to be assessed?

The criteria for assessing leadership in the field of cybersecurity is for future scholars to create an interdisciplinary conceptual framework to investigate behavioral cybersecurity, human factors, modelling and simulation. The scholars should also incorporate enterprises both private and public in research to ensure that the projected results will produce the intended results. The scholar gives practical recommendation that employees be trained and educated on human factors associated with cyberse-curity. Also, the researcher recommended training college and university students on human fac-tors and how they associate with cybersecurity threats.

3.The article should include future research directions and/or recommendations for the improvement of cybersecurity leadership.

The future research study employed a systematic review of literature that explored how the different human factors influenced digital security of organizations. While, systematic review of literature presents an opportunity to explore the area under study from a wider perspective, it is limited with time and accuracy of pre-sented data and size of the sample used. One common limitation associated with systematic review is that it is not easy to verify the authenticity and accuracy of information and results provided. The second limitation is that while it is an easy research method, it is highly time consuming and tedious. However, the findings provided in this study sets ground for future work on human factors and how they facilitate unintentional cyberthreats and cyberattacks. For future work, re-searchers should consider qualitative research to understand unintentional cyber-attacks and human factors from the perspectives and views of participants.

Round 2

Reviewer 1 Report

I still don't see much significance in advancing the field with this paper. The conclusion that humans are still the weakest links can be based on simple observation an actions by analyzing attack vectors. A more in-depth analysis is needed to study tactics techniques and procedures corresponding to human behavior in early stages of attack vectors. I am not convinced that this paper is journal-ready.

Author Response

Response to Reviewer 1 Comments

Point 1: I still don't see much significance in advancing the field with this paper. The conclusion that humans are still the weakest links can be based on simple observation an actions by analyzing attack vectors. A more in-depth analysis is needed to study tactics techniques and procedures corresponding to human behavior in early stages of attack vectors. I am not convinced that this paper is journal-ready.

Response 1: 

The conclusion that humans are still the weakest links can be based on simple observation an actions by analyzing attack vectors. A more in-depth analysis is needed to study tactics techniques and procedures corresponding to human behavior in early stages of attack vectors. In the midst of growing cybersecurity attacks human factors are leading attack vectors to cyber-incidents.   Cybersecurity leadership is vital to managing and controlling the threat of widespread attacks.  Imagining that cyber security expertise and abilities are critical in developing an operational organization. Specifically, the individuals who do not have a security background needs to be experienced and proficient in recognizing the act of inappropriate behavior. Targeted cyber-attacks are successful due to incidents, like inadequate security experience, carelessness, and distortion.  Technology only may not be enough to impose the needed protection in the system if the human elements fail to detect and sustain policies in the system of security. It is simply through training the workforce properly operating, will help to alleviate the threat of this specific attack vector (Ani et al., 2019).

Reviewer 2 Report

Thank you for the opportunity to provide a review report to the manuscript.

The paper was improved as compared to the original version but still lack the following issues, which some of them were not taken although requested in my previous review report.

I suggest the authors to re-read the comments point by point and address them in the second-round revision of the manuscript.

Minor Comments for the Authors:

  1. Please include a reference to the repository (Such as Google Scholar; WoS; NCBI) in addition to the following attributes: citation Purpose Participants Method Findings Suggestions/
  2. Table 1: Please try to shorten the findings into a short paragraph. (See for example: “”5% of cyber-related attacks on businesses are propagated by human factors. For instance, increasing attacks, ransomware attacks and data breaches are propagated by human factors. Despite the knowledge, the scholars showed that managers were still reluctant to equip themselves with the knowledge, skills, and expertise to effective mitigation against cyberattack”. Such long text should be shortened and not being cut and pasted into the table from the affiliated paper. In addition, if you cut and paste a text directly from the paper, please put the text in “ “ and also include a ref to the journal and a page number from where the citation was taken from. In addition, please include a ref to Table 1 & Table 2 in the text.
  3. The ref of ZZwilling et al. (2022) is not correct cited, please correct to Zwilling et al. (2022), please also check all references and make sure they are correclty cited according to the journal’s guidelines.
  4. In the previous review report I wrote:: “Please provide a contribution part of the findings to the explored domain as part of the discussion or as a separate part.” This request was not being addressed by the authors. Please provide a paragraph which summarizes the discussion part. You can start your paragraph by mentioning: “To summarize the discussion section, we find that the current study with its findings contribute to the explored domain with the following: … 1. XXX; 2. XXX etc.
  5. In my previous round revision, I suggested the authors to exhibit the most prominent papers in the explored domain that are evaluated in the table and include the following details with respect to them in a separate table: Year of publication, Authors, Journal (J) or Conference (C), Title, #of citations. The authors answered: “The most prominent papers have been placed in the table with the Year of publication, Authors, and Citation”. However, the authors did not provide such table in this revision. It is very disappointing to request the same issue again, while the authors claim that they provided the requested response. I recall the authors to provide the suggested table.
  6. In my previous revision I requested the authors to provide the following: “The discussion doesn’t address the issue of organizations with or without security policy and enforcement of regulations (in case the firm is regulated). It is suggested to add a column in the table and include in the discussion an attitude related to enforcement and enforcement of regulations by the firm as related to awareness of employees to cyber security if applicable.”

The author’s response was: “The suggestions were added to a table or column included in the discussion an attitude related to enforcement and enforcement of regulations by the firm as related to awareness of employees to cyber security if applicable. Organizational leaders are complacent or unintentional in instituting policies and measures that would protect organizations from cyberattacks. Organizations have 4 become reluctant in training employees on cybersecurity increasing the organizations’ vulnerability”. Unfortunately, the authors didn’t provide any column in the table that indicates weather enforcement of regulation is applicable or not, I would like to see such a column in the revised version.
